# Trends in the Epidemiology of *Pneumocystis* Pneumonia in Immunocompromised Patients without HIV Infection

**DOI:** 10.3390/jof9080812

**Published:** 2023-07-31

**Authors:** Ting Xue, Xiaomei Kong, Liang Ma

**Affiliations:** 1NHC Key Laboratory of Pneumoconiosis, Key Laboratory of Prophylaxis and Treatment and Basic Research of Respiratory Diseases of Shanxi Province, Shanxi Province Key Laboratory of Respiratory, Department of Respiratory and Critical Care Medicine, First Hospital of Shanxi Medical University, Taiyuan 030001, China; 2Critical Care Medicine Department, NIH Clinical Center, Bethesda, MD 20892, USA

**Keywords:** *Pneumocystis*, non-HIV/AIDS, immunocompromised, epidemiology, primary immunodeficiency, immunodepleting monoclonal antibodies, organ transplantation

## Abstract

The increasing morbidity and mortality of life-threatening *Pneumocystis* pneumonia (PCP) in immunocompromised people poses a global concern, prompting the World Health Organization to list it as one of the 19 priority invasive fungal diseases, calling for increased research and public health action. In response to this initiative, we provide this review on the epidemiology of PCP in non-HIV patients with various immunodeficient conditions, including the use of immunosuppressive agents, cancer therapies, solid organ and stem cell transplantation, autoimmune and inflammatory diseases, inherited or primary immunodeficiencies, and COVID-19. Special attention is given to the molecular epidemiology of PCP outbreaks in solid organ transplant recipients; the risk of PCP associated with the increasing use of immunodepleting monoclonal antibodies and a wide range of genetic defects causing primary immunodeficiency; the trend of concurrent infection of PCP in COVID-19; the prevalence of colonization; and the rising evidence supporting de novo infection rather than reactivation of latent infection in the pathogenesis of PCP. Additionally, we provide a concise discussion of the varying effects of different immunodeficient conditions on distinct components of the immune system. The objective of this review is to increase awareness and knowledge of PCP in non-HIV patients, thereby improving the early identification and treatment of patients susceptible to PCP.

## 1. Introduction

*Pneumocystis jirovecii* is an atypical opportunistic fungus that can cause life-threatening *Pneumocystis* pneumonia (PCP) in immunodeficient patients. Prior to the advent of the HIV/AIDS epidemic, PCP was predominantly reported in premature or malnourished infants and patients with cellular immunodeficiencies due to conditions such as hematological malignancies, T cell deficiency, and other severe diseases requiring corticosteroid treatment [1,2]. In the 1980s, the high morbidity of PCP in HIV-infected patients overshadowed its prevalence in non-HIV patients. Since the late 1990s, the broad utilization of prophylaxis and subsequent combination antiretroviral therapy (cART) has significantly reduced the PCP incidence in HIV-infected patients, but PCP has continued to affect individuals with other immunodeficient conditions [3,4]. In fact, some studies have reported an increase in PCP incidence among these patients [5,6]. It has been estimated recently that healthcare associated with PCP in the United States alone costs $475–$686 million annually, ranking among the top three or four most serious fungal diseases [7,8,9]. 

Compared to PCP in HIV-infected patients, PCP in a non-HIV immunosuppressed population has a shorter incubation period, a higher risk of respiratory failure and mortality, and a more rapid disease progression, making it more difficult to diagnose in a timely manner [10,11,12,13,14]. Given the serious threat PCP poses to immunodeficient patients, it is important for clinicians and health care workers to be aware of the new trends in the epidemiology of PCP and to identify potential susceptible patients at the earlier stage in order to initiate appropriate and timely prophylaxis or treatment. 

Despite the growing public health implications of PCP, this and other invasive fungal diseases have received little attention and few resources, with limited access to quality diagnostics and treatment. This situation has recently prompted the World Health Organization (WHO) to list PCP as one of the 19 priority invasive fungal diseases, calling for increased research efforts and public health action [15]. In response to this important initiative, we contributed this review, with a focus on the changing epidemiology of PCP among non-HIV patients. 

## 2. The Infectious Agent and Infection Chain of PCP

*P. jirovecii*, the pathogen responsible for PCP in humans, belongs to the genus *Pneumocystis,* which is closely related to the genera *Schizosaccharomyces*, *Taphrina*, and *Saitoella* of the subphylum Taphrinomycotina within the phylum Ascomycota phylum [16]. This genus encompasses a large group of phylogenetically closely related yet clearly distinct organisms that are potentially capable of infecting all mammalian species with a very strong, though probably not strict, host specificity. Currently, only six *Pneumocystis* species have been formally named, including *P. jirovecii* (infecting humans or *Homo sapiens*) [17,18], *P. murina* (house mice or *Mus musculus*) [19], *P. carinii* (Brown or Norway rats or *Rattus norvegicus*) [20], *P. wakefieldiae* (Brown or Norway rats or *Rattus norvegicus*) [21], *P. oryctolagi* (rabbits or *Oryctolagus cuniculus*) [22], and *P. canis* (dogs or *Canis lupus familiaris*) [23]. Long-term cultivation of *Pneumocystis* remains unsuccessful, which impedes research on many aspects of this pathogen, particularly its life cycle, transmission, immune evasion, and drug resistance and susceptibility.

Despite great difficulty in studying *Pneumocystis*, recent advances in genetics have facilitated our understanding of the basic biology and epidemiology of this pathogen, including its chain of infection, as summarized in Figure 1.

Studies with animal models have confirmed that cysts (also known as asci) serve as the infectious agent [24]. Other life stages, including the trophic form, presumably cannot serve as the infectious agent due largely to the lack of a protective cell wall, which prevents their survival, even for a short transient period, in a natural environment outside the lungs of the host [16]. It is now evident that human infection with *P. jirovecii* is not of zoonotic origin, as strongly supported by the pathogen’s high-degree of host-species specificity and consistent failure in experimental cross-species infections.

No natural environmental reservoir has been identified, while there is overwhelming evidence indicating that humans themselves serve as carriers or reservoirs of this pathogen [16]. Potential human reservoirs include PCP patients and individuals, including children (particularly infants) and adults with either weakened or competent immunity, who are either colonized or have subclinical infection with *P. jirovecii*. There is no doubt that the respiratory tract serves as both the portal of exit and entry for *Pneumocystis* infection. The mode of transmission is now generally believed to be airborne through person-to-person contact. Efficient transmission likely requires proximity to an infected host––for example, in the same room [25,26]. 

PCP primarily affect individuals with impaired immunity. Susceptible individuals include those with various congenital or acquired immunodeficiencies, as detailed below. Excluded from this review are the acquired immunodeficiencies caused by HIV and other retroviruses, including human T-lymphotrophic virus (HTLV). 

## 3. Increased Incidence of PCP in Patients without HIV/AIDS

Over the past two decades, there has been an increase in the incidence of PCP in non-HIV patients based on large-scale national studies. According to the national surveillance data in England, the annual PCP incidence in non-HIV patients rose from 3.15 cases per million general population between 2000 and 2005 to 5.13 cases between 2006 and 2010 [27]. In France, a study of 604 PCP cases in a single hospital from 2005 to 2013 showed an increase in the proportion of PCP cases from 63% to 85% in non-HIV patients and a decrease from 37% to 15% in HIV-infected patients [14]. In Germany, a multi-center, longitudinal population-base study of 12,455 PCP cases from 2014 to 2019 (including 2,124 HIV-related and 10,331 non-HIV-related) showed a significant decrease in annual HIV-related PCP cases from 346 to 331, while the number of non-HIV-related PCP cases increased substantially from 1511 to 1841 [28].

The increase in PCP incidence is attributed to a variety of factors, including the more aggressive use of immunosuppressive agents, increased implementation of organ and stem cell transplantation, application of more sensitive diagnostic methods, and improved awareness of PCP by physicians (reviewed in references [23,29,30,31,32,33]. The true incidence of PCP at present is difficult to determine due to the increased use of prophylaxis, which may lead to underestimation of the actual incidence.

## 4. Immunodeficient Conditions and Risk Factors for PCP

### 4.1. Immunosuppressive Agents

Various immunosuppressive or immunomodulatory agents have long been recognized as important risk factors for the development of PCP in non-HIV patients (Table 1). These agents can cause significant immunosuppression, primarily of cellular immunity, which is thought to be the major defense against *Pneumocystis*. Among the most significant immunosuppressive agents contributing to the occurrence of PCP in non-HIV patients are corticosteroids, which are widely used to treat various underlying conditions, including cancers, autoimmune and inflammatory diseases, and organ and stem cell transplantation, as described below. Several studies have shown that ~90% of PCP patients without HIV infection received corticosteroid therapy, usually in combination with other immunosuppressive agents [3,4,34]. Initially, only long-term and high-dose corticosteroid use was identified as a high risk for PCP, but recent studies suggest that intermittent corticosteroid therapy is also an important risk factor for developing PCP [35]. However, underlying conditions are a more important risk factor for PCP; for example, asthma patients (usually not immunocompromised) treated with steroids are typically not at risk for PCP.

In addition to corticosteroids, traditional immunosuppressive or chemotherapeutic agents associated with the development of PCP include various antimetabolites (methotrexate), alkylating agents (cyclophosphamide), anticalcineurins (cyclosporine and tacrolimus), and inhibitors of a mammalian target of rapamycin (mTOR) (sirolimus and everolimus) (Table 1). Beginning in the 1970s, these chemotherapeutic agents have been frequently reported as risk factors predisposing patients to PCP following their use to treat underlying disorders, primarily hematological malignancies, solid organ transplantation rejection, and rheumatoid arthritis. Subsequently, these drugs were gradually extended to other underlying disease conditions, thereby placing new disease categories at risk for PCP. Since the early 2000s, new immunotherapeutic agents that are utilized in the treatment of various diseases, particularly a wide array of monoclonal antibodies targeting TNFα and other cytokines [36] as well as lymphocyte antigens, have been further recognized as risk factors for PCP (Table 1). 

**Table 1 jof-09-00812-t001:** Drugs associated with the development of PCP.

Drugs	Mechanisms of Action	Chemical Properties	Approved Applications	References *
Corticosteroids **	Suppression of inflammation, leukocyte migration and activation; induction of apoptosis	Steroid hormones	Various diseases	[4,32,37]
*Inhibitors of DNA/RNA synthesis*				
Temozolomide	Inhibit DNA and cellular replication	Alkylating agent/imidazotetrazine	Brain cancer, astrocytoma, and glioblastoma multiforme	[38,39]
Cyclophosphamide	Inhibit DNA and cellular replication	Alkylating agent/phosphoramide mustard	Lymphoma, multiple myeloma, leukemia, ovarian cancer, breast cancer, small cell lung cancer, neuroblastoma, and sarcoma; organ transplant rejection	[40,41]
Bleomycin	Induce DNA strand breaks	Nonribosomal peptide	Lymphoma, testicular cancer, ovarian cancer, and cervical cancer	[42,43]
Fluorouracil	DNA synthesis inhibitor	Antimetabolite/pyrimidine analog	Various cancers	[43,44]
Cytarabine	DNA synthesis inhibitor	Antimetabolite	Leukemia and lymphoma	[43,45]
Methotrexate	DNA/RNA synthesis inhibitor	Antimetabolite/antifolate	Cancers, autoimmune diseases, and ectopic pregnancy	[43,46]
Azathioprine	Purine synthesis inhibitor	Antimetabolite/Purine analog	Rheumatoid arthritis, Crohn’s disease, ulcerative colitis, and kidney transplant rejection	[43,47,48]
Cladribine	Purine synthesis inhibitor	Antimetabolite/Purine analog	Leukemia and lymphoma	[43,49]
Fludarabine	Purine synthesis inhibitor	Antimetabolite/Purine analog	Leukemia and lymphoma	[43,50,51,52]
*Inhibitors of immune functions*				
Rituximab	B-cell signaling inhibitor	Anti-CD20 monoclonal antibody	Autoimmune diseases, Hematological cancers	[53,54]
Alemtuzumab	Deplete CD52-bearing B and T cells	Anti-CD52 monoclonal antibody	Hematological cancers, multiple sclerosis, and organ transplant rejection	[43,52,55]
Abatacept	T-cell signaling inhibitor	Recombinant protein	Rheumatoid arthritis, juvenile idiopathic arthritis, and psoriatic arthritis	[56,57]
Belatacept	T-cell signaling inhibitor	Recombinant protein	Organ transplant rejection	[58,59,60]
Tocilizumab	Anti–IL-6 receptor	Anti-IL6 receptor monoclonal antibody	Rheumatoid arthritis, juvenile rheumatoid arthritis	[61,62,63]
Adalimumab	TNFα inhibitor	Anti-TNFα monoclonal antibody	Rheumatoid arthritis, psoriatic arthritis, ankylosing spondylitis, Crohn’s disease, ulcerative colitis, chronic psoriasis, hidradenitis suppurativa, and juvenile idiopathic arthritis	[64,65]
Etanercept	TNFα inhibitor	Recombinant protein	Rheumatoid arthritis, juvenile rheumatoid arthritis and psoriatic arthritis, plaque psoriasis and ankylosing spondylitis	[43,66,67]
Golimumab	TNFα inhibitor	Anti-TNFα monoclonal antibody	Rheumatoid arthritis, psoriatic arthritis, ankylosing spondylitis, ulcerative colitis, and rheumatoid arthritis	[68,69]
Infliximab	TNFα inhibitor	Anti-TNFα monoclonal antibody	Crohn’s disease, ulcerative colitis, psoriasis, psoriatic arthritis, and ankylosing spondylitis	[43,70,71]
Cyclosporine	Calcineurin inhibitor	Anticalcineurin	Autoimmune diseases, and organ transplant rejection	[43,72,73,74]
Tacrolimus	Calcineurin inhibitor	Anticalcineurin/macrolide lactone	Organ transplant rejection, eczema, uveitis, and vitiligo	[75,76,77]
Everolimus	Inhibitor of mammalian target of rapamycin (mTOR)	Derivative of sirolimus	Organ transplant rejection, kidney cancer, breast cancer, and subependymal giant cell astrocytoma	[78,79,80]
Sirolimus (rapamycin)	Inhibitor of mammalian target of rapamycin (mTOR)	Macrolide compound	Organ transplant rejection, lymphangioleiomyomatosis	[43,81,82,83]

* For the association of the drug with the risk of PCP. ** Including prednisone, prednisolone, dexamethasone, hydrocortisone, etc.

Of special note is the risk of PCP associated with the growing use of a class of monoclonal antibodies targeting the CD20 antigen on B lymphocytes, including rituximab, ofatumumab, obinutuzumab, veltuzumab, and ocrelizumab. Among them, rituximab is the first generation of this class (approved in 1997), and it is frequently used to treat certain lymphomas and leukemias as well as various autoimmune diseases. It is the first known immunosuppressive agent associated with the development of PCP resulting primarily from suppression of B cell function, as has been documented in more than 100 reports worldwide [35,53,54,84,85], while there are only sparse reports of PCP in patients receiving the newer generation of anti-CD20 agents [86]. A range of 1.5–13% incidence rates of PCP has been reported in patients treated with rituximab for various underlying diseases, such as hematologic malignancies, rheumatic diseases, and solid organ transplantation [53,84,85,87]. Among these studies is one showing a 10% incidence of PCP in patients with hematologic malignancies who were treated with rituximab but without concomitant chemotherapy or significant glucocorticoid exposure [53]. These high incidences, together with the results of animal studies [88], support the important role of B cells in the host’s immune defense against PCP. 

There are multiple monoclonal antibody agents used to suppress organ transplantation rejection that can increase the risk of PCP, including alemtuzumab, belatacept, cyclosporine, tacrolimus, everolimus, and sirolimus [52,55,58,89,90]. There are also multiple monoclonal antibody agents used to treat rheumatoid arthritis that can increase the risk of PCP, including abatacept, tocilizumab, adalimumab, etanercept, and golimumab.

### 4.2. Cancers 

Cancers are one of the earliest identified underlying conditions associated with PCP, as first reported in late 1950s in patients with leukemia and Hodgkin’s disease [91,92]. Hematological malignancies (including various leukaemias and lymphomas as well as multiple myeloma) remain the most common underlying diseases predisposing patients to PCP, accounting for ~30–80% of PCP cases among non-HIV patients [10,29,31]. Solid tumors are also closely associated with the development of PCP among non-HIV patients, accounting for 7.9–26% of cases [93,94,95,96,97,98,99,100]. A single-center retrospective study of PCP in solid tumors in Japan showed lung cancer to be the most common underlying tumor (30%), followed by breast cancer (15%) [101]. 

Multiple factors may contribute to the development of PCP in these cancer patients, including cancer-induced immune dysfunction, chronic use of corticosteroid therapy and other chemotherapies and immunotherapies, and radiation therapy [101,102]. There are studies demonstrating that PCP can be effectively prevented by chemoprophylaxis, as evidenced by the absence of PCP in 80 high-risk patients receiving trimethoprim-sulfamethoxazole compared to the occurrence of PCP in 21% (17/80) of patients receiving a placebo [103].

### 4.3. Solid Organ and Stem Cell Transplantation

The first case of PCP in a transplant recipient was reported in 1964 from the United States in a boy following renal transplantation [47]. A search of PubMed using keywords “*Pneumocystis* AND Transplant” in “Title/Abstract” resulted in 751 articles from 1964 to 2022, with 75% of them published over the past two decades (Figure 2). Most of the recent cases were reported from outbreaks, as discussed below in Section 5. 

PCP is an important and dangerous cause of morbidity and mortality in transplant recipients, especially those with delayed diagnosis or no prophylaxis. Studies of transplant recipients without prophylaxis have reported a PCP incidence of 5–16%, depending on the type of transplanted organ, the transplant center, and the immunosuppressive regimens utilized (reviewed in references [10,104]). The highest incidences (10–40%) occurred among lung, combined lung-heart, and kidney transplant recipients [105,106,107]. Without prophylaxis, the first 6 months after solid organ transplantation have the highest risk for developing PCP, presumably due to intensified immune suppression to prevent graft rejection during this period. Nonetheless, PCP cases may still occur beyond 12 months after transplantation despite a full course of prophylaxis [108]. 

The occurrence of PCP after transplantation is associated with many additional risk factors, such as older ages, concomitant cytomegalovirus (CMV) infection, low CD4 cell counts, hypogammaglobulinemia, BK polyomavirus-related diseases, and human leukocyte antigen mismatches, as has been extensively reviewed [109,110,111]. 

### 4.4. Autoimmune and Inflammatory Diseases 

PCP has been reported as a complication of a great variety of autoimmune and inflammatory diseases (AIIDs), including at least 19 individual diseases, as listed in Appendix A [4,40,70,97,112,113,114,115,116,117,118,119,120,121,122,123,124,125,126,127,128,129,130,131,132,133,134,135,136,137,138,139,140,141,142]. Among them, the most frequently reported diseases associated with PCP are granulomatosis with polyangiitis (GPA, formerly known as Wegener’s granulomatosis), systemic lupus erythematosus (SLE), rheumatoid arthritis (RA), and inflammatory bowel disease (IBD, including Crohn’s disease and ulcerative colitis) [32,123,143]. Some AIIDs are caused by inherited genetic defects, as discussed below in the Primary Immunodeficiencies section.

GPA is a rare necrotizing vasculitis characterized by necrotizing granulomas and pauci-immune vasculitis that most commonly affects small- to medium-sized vessels in the respiratory tract and kidneys. The first case of PCP associated with GPA was reported in 1960 from a female American adult [92]. Based on multiple systematic reviews of data reported worldwide, GPA appears to be the most common AIID associated with PCP, with an estimated incidence of 6–20% [32,144,145]. The frequency of PCP in GPA in the United States and France is estimated to be 60–89 and 120 cases per 10,000 patients per year, respectively [114,124,146]. Godeau et al. [124] reported that GPA is more commonly associated with PCP (35%) than SLE (18%) and dermatomyositis (15%) among AIIDs. The main risk factors for developing PCP in GPA patients include the use of glucocorticoids and other immunosuppressive agents (such as cyclophosphamide and infliximab). 

SLE patients are immunodeficient as part of their disease pathogenesis and as a result of their treatment, and are thus prone to opportunistic infections. The occurrence of PCP in SLE patients has been frequently reported since 1975 [147]. Based on the most recent review of 18 large-scale studies worldwide, the overall incidence of PCP in SLE patients is estimated to be 0.17% [148]. Most of these patients received high-dose corticosteroids and other chemotherapeutic and immunotherapeutic agents, and their disease course was complicated by lupus nephritis and lymphopenia, with low CD4 cell counts [122,149]. Pulmonary infections, especially CMV and *Pneumocystis*, are closely associated with increased mortality in SLE patients. 

Based on reports from the 1990s from France [124] and the United States [114], the incidence of PCP among patients with RA was estimated to be 1–2 cases per 10,000 patients/year. Recent reports have shown a similar incidence in the UK [112] and a decrease incidence (0.6–4.0 PCP cases/100,000 RA patients/year) in the United States [113]. In contrast, a higher incidence of PCP has been reported from Japan among patients with RA, especially those receiving various newly developed immunotherapeutic agents, including TNFα inhibitors, T cell signaling inhibitors, and anti-IL-6 receptor antibody, with an incidence of 0.1–0.4% and mortality of 10.1–22.2% [115,116,117,150]. PCP incidence in RA is associated with multiple risk factors, such as older age, male sex, coexisting lung diseases, and therapy with methotrexate and corticosteroids [116,118,119]. Additionally, several studies have shown a relatively normal lymphocyte and CD4 cell counts in RA patients with PCP, suggesting the possibility of a different mechanism than T cell deficiency in the development of PCP in these patients [120,151].

IBD represents a group of intestinal disorders that cause chronic inflammation of the digestive tract. The two most common types of IBD are Crohn’s disease and ulcerative colitis. Almost all PCP cases in IBD patients were reported after the 1990s. The reported incidence of PCP in IBD varies from 0.03% [140] to 0.3% [136]. Based on the most recent review by Lawrence et al. [137], there have been a total of ~90 PCP cases in IBD patients reported from ~30 studies across the world. Among these patients, 88% received corticosteroids as IBD treatment, 44% received TNF inhibitors, 42% received thiopurines, and 15% received cyclosporine or tacrolimus. While CD4 cell counts were not available for most of these patients, 86% of them were lymphopenic. Once PCP has developed, the mortality rate of patients with IBD is high (18%) and strongly associated with low serum albumin levels [139]. 

### 4.5. Primary or Congenital Immunodeficiencies in Children 

PCP is a significant cause of morbidity and mortality in immunocompromised children. In fact, it was the epidemic of interstitial plasma cell pneumonia in premature infants and malnourished children in the mid-20th century in Europe that led to the first recognition of *Pneumocystis* as a human pathogen [152,153]. Infants are at a higher risk for infection than healthy older children and adults because their immune systems are not fully developed, with limited or weakened innate and adaptive immune responses to pathogens [154,155]. Premature birth and malnourishment, along with the use of corticosteroids and other immunosuppressive agents as part of their treatment, can further compromise the immune system and increase their susceptibility to infection, including *P. jirovecii* infection [153,156,157,158,159,160].

The high susceptibility of infants and children to *P. jirovecii* is supported by the high seroprevalence of anti-*Pneumocystis* antibodies in healthy children early in life [161,162], and the high prevalence (100%) of *P. jirovecii* colonization detected by PCR in immunocompetent infants [163]. 

Various primary or hereditary immunodeficient diseases caused by genetic defects in germline variants in single genes can also increase the risk of PCP in children. In some early studies, primary immunodeficiency constitutes the most frequent underlying diseases associated with PCP in infants [164]. We provided a comprehensive list of primary immunodeficient diseases associated with PCP (Table 2 and Appendix A [86,165,166,167,168,169,170,171,172,173,174,175,176,177,178,179,180,181,182,183,184,185,186,187,188,189,190,191,192,193,194,195,196,197,198,199,200,201,202,203,204,205,206,207,208,209,210,211,212,213,214,215,216,217,218,219]). There have been a total of 485 primary immunodeficient diseases or inborn errors of immunity (IEI) recognized by the International Union of Immunological Societies Expert Committee, which are classified into 10 major categories, including immunodeficiencies affecting cellular and humoral immunity, combined immunodeficiencies with associated or syndromic features, predominantly antibody deficiencies, diseases of immune dysregulation, congenital defects of phagocyte number or function, defects in intrinsic and innate immunity, autoinflammatory disorders, complement deficiencies, bone marrow failure, and phenocopies of inborn errors of immunity [220]. Out of these 485 diseases, 44 from nine categories (without the last one) have been associated with PCP, including 13 (30%) from immunodeficiencies affecting cellular and humoral immunity, and another 13 (30%) from combined immunodeficiencies with associated or syndromic features. More than half of the 44 diseases associated with PCP were reported in the past 5 years (Appendix A). While individually rare, the cumulative number of individuals with these inherited diseases poses a significant health burden due to their prevalence of 1–5 cases per 1000 people and an annual increase in ~20 new types of genetic defect [220]. Children with these diseases may require long-term immunosuppressive therapy or stem cell transplantation, which can further increase their risk of developing PCP. 

A serial cross-sectional study of US pediatric hospital discharges (aged 0–18 years) from 1997–2012 identified 1902 PCP cases. Cases with HIV infection decreased from 285 in 1997 to 29 cases in 2012, whereas hematologic malignancy and primary immunodeficiency became more prominent. All-cause in-hospital mortality was 12% and was particularly high among cases with hematopoietic stem cell transplant (32%) [221]. 

A recent single-center study of hospitalized children (<3 years old) in France found that 11% (32/279) of them were infected with *P. jirovecii*, with the highest positivity rate found in children with cardiopulmonary pathologies (22%), followed by SCID (19%), hyaline membrane disease (16%), asthma (9%), and acute leukaemia (6%) [222]. Of the infected children, 40% were considered colonization cases while 56% had confirmed PCP, mainly associated with SCID and other congenital diseases.

### 4.6. COVID-19 

COVID-19, caused by the severe acute respiratory syndrome coronavirus 2 (SARS-CoV-2), can weaken the host’s immune system and increase the risk of secondary infections with bacterial and fungal pathogens, including *P. jirovecii* [223,224,225]. There have been approximately 100 reports of coinfection of *P. jirovecii* in COVID-19 patients, with some of them reviewed recently by Gioia et al. [226] and Khodadadi et al. [227]. The largest number of coinfection cases (in non-HIV/AIDS) was reported from France (34 cases), followed by Italy (22 cases) and Pakistan (10 cases) (Figure 3 and Appendix A [15,224,225,226,228,229,230,231,232,233,234,235,236,237,238,239,240,241,242,243,244,245,246,247,248,249,250,251,252,253,254,255,256,257,258,259,260,261,262,263,264,265,266,267,268,269,270,271]).

While there is evidence of significant depletion of CD4 T cells in COVID-19 patients [272] and CD4 cell deficiency is the most important driving factor for the development of PCP, as seen in AIDS, the incidence of PCP in COVID-19 appears to be far lower than that reported in AIDS. This may reflect the differences in T cell deficiency between COVID-19 and AIDS. AIDS is a chronic disease; its T cell deficiency is usually progressive and irreversible if untreated. In contrast, COVID-19 is primarily an acute illness; its T cell deficiency is generally temporary, reversible, and less severe, and, in most cases, T cell numbers and function improve as the patient recovers from the infection [273,274,275,276,277,278,279]. Another potential difference is that AIDS primarily targets CD4 T cells while COVID-19 appears to affect CD8 T cells more than CD4 T cells in mild cases of COVID-19, as evidenced by a greater reduction or cytotoxic response of CD8 T cells compared with CD4 T cells in these cases in several studies [280,281,282,283], though this trend was not supported by other studies [284,285]. 

While it is clear that the incidence of PCP is much lower in COVID-19 than that in AIDS, the incidence of PCP in COVID-19 seems to be underestimated for various reasons [223,226,286], including the difficulties in distinguishing between these two diseases due to their similarities in clinical symptoms and manifestations, the lack of awareness or attention to PCP, and the lack of access to sensitive diagnostic tests for PCP, particularly during the challenging pandemic era. A study using a highly sensitive qPCR assay identified a prevalence of 9% of *P. jirovecii* infection in critically ill COVID-19 patients [228], supporting the possibility of a high prevalence of PCP in COVID-19 patients and highlighting the importance of using a sensitive, pathogen-specific diagnostic method. Early diagnosis of PCP is crucial for the timely treatment of patients because coinfection with PCP may increase the severity and mortality of COVID-19 [286,287].

There are multiple potential risk factors for the development of PCP in COVID-19, including cytokine storm, damage to the airway epithelial barrier, mechanical ventilation, admissions to intensive care units (ICU), use of glucocorticoids and other immunosuppressive agents, underlying diseases (pulmonary diseases, acute respiratory distress syndrome, and kidney disease), and prolonged hospitalization [224,227,287]. Nonetheless, in some cases, it may be challenging to determine whether the onset of PCP in COVID-19 patients is solely due to COVID-19-induced immune defects, the administration of immunosuppressive agents during COVID-19 treatment, or underlying conditions prior to COVID-19 infection [288]. Further research is needed to better understand the role of these risk factors and thus help to develop appropriate and effective strategies to prevent or reduce the risk of PCP in COVID-19, ultimately improving patient outcomes.

### 4.7. Other Underlying Conditions 

In addition to the underlying diseases described above, other conditions associated with a high risk of *P. jirovecii* infection, though not necessarily PCP, include chronic lung diseases, nephrotic diseases, diabetes mellitus, dermatologic diseases, CMV infection, BK polyomavirus-related infections, and surgical operation [109,289,290,291]. The prevalence of *P. jirovecii* in these conditions is highly variable depending on the patient populations. A large-scale national study of 4293 PCP patients in Japan has revealed diabetes mellitus to be the second most common underlying condition (30%) after hematologic malignancy (31%) for PCP in adults [291]. Patients with various chronic lung diseases, such as chronic obstructive pulmonary disease (COPD), interstitial lung disease (ILD), severe asthma (SA), and cystic fibrosis (CF) have a high prevalence of *Pneumocystis*, with most of them presumably representing colonization rather than active infection, as further discussed below.

### 4.8. Effects of Different Immunocompromised Conditions on Immune Functions 

The host immune response plays a vital role in the development of PCP, which entails intricate interactions among numerous components, as has been extensively reviewed in recent publications by Charpentier et al. [292] and Otieno-Odhiambo et al. [293]. Immunocompetent individuals are capable of controlling and clearing *Pneumocystis* without experiencing symptoms, whereas immunodeficient individuals may develop severe and potentially life-threatening pneumonia. The risk for PCP depends on the interplay between epidemiologic exposure to *P. jirovecii* and the nature and severity of specific immunocompromised conditions. Immunocompromised conditions vary in their effects on specific components of the immune system and in the degree to which they impact its functioning. Some conditions have a broad impact on the entire immune system, while others selectively affect only particular components. 

In early reported sporadic PCP cases associated with immunodeficient diseases, humoral immunodeficiency is observed more frequently than cellular immunodeficiency, leading to speculation that humoral immunodeficiency plays a primary role in driving the occurrence of PCP [294]. However, the remarkably high incidence of PCP outbreaks during the AIDS epidemic in 1980s, together with evidence from animal models, underscores the paramount significance of cellular immunodeficiency (particularly CD4 T cell deficiency) in the pathogenesis and development of PCP [292,293]. Patients with CD4 T cell counts lower than 200 cells/µL have a high risk for PCP [295].

Among all known non-HIV/AIDS immunocompromised conditions, glucocorticoid therapy, widely utilized in the treatment of almost all immunocompromised diseases, stands out as the predominant risk factor for PCP. Glucocorticoids inhibit both cellular and humoral immunity as well as innate immunity [296], so they appear to play a triple role in predisposing patients to the development of PCP. Immunodepleting monoclonal antibody agents, increasingly employed to treat cancers and AIIDs, have variable impacts on specific components of the immune system depending on their specific target; they may predominantly impair innate immunity (e.g., eculizumab and etanercept), cellular immunity (alemtuzumab, abatacept, and belatacept) or humoral immunity (rituximab and Epratuzumab), respectively [297,298]. Most of the chemotherapeutic agents used to treat cancers and AIIDs primarily suppress the production of white blood cells, particularly neutrophils, which are an essential component of innate immunity. Approximately half of the cancer patients undergoing chemotherapy develop neutropenia [299]. Neutropenia can lead to a weakened immune system and increased susceptibility to *P. jirovecii* infection [300]. 

For cancer patients, in addition to the immunosuppressive and chemotherapeutic agents discussed above, radiation therapy, especially when targeting the bone marrow or the lymph nodes, can damage all components of the immune system and impair their ability to function effectively [301]. Furthermore, many cancers, particularly blood cancers (e.g., leukemia, lymphoma), can cause extensive disruption of haematopoiesis and directly compromise the immune response [302,303]. Cancer cells may evade immune detection or inhibit immune cell function [304], thus impairing the body’s ability to mount an effective immune response against infections. 

All organ transplant recipients require the use of immunosuppressive agents, which have varying effects on different components of the immune system, as discussed above. In solid organ transplantation, the primary goal of immune suppression is to suppress the T cell response and thus prevent organ rejection [305]. In stem cell transplantation, immune suppression usually targets both T cells and other components of the immune system, including B cells and natural killer cells, to prevent rejection of the transplanted stem cells and reduce the risk of graft-versus-host disease. Immune suppression in solid organ transplantation is generally more intensive and long-term compared to stem cell transplantation. As immunosuppression leaves the transplant recipients susceptible to various infections, success in transplantation requires careful management of the balance between the risks of graft rejection and the increased susceptibility to various infections, including PCP [289]. 

Primary immunodeficiency can affect all components of the immune system. More than half (60%) of all known primary immunodeficient diseases associated with PCP involve impairment of both cellular and humoral immunity (Appendix A). Less than 10% of them predominantly affect humoral or innate immunity alone.

COVID-19 primarily targets the respiratory tract, leading to lung tissue damage and triggering overexaggerated and dysregulated pro-inflammatory and inflammatory responses. These effects extend beyond the innate defense system and have a broad impact on the adaptive immune system [306]. Patients with severe COVID-19 have marked reductions in the number of CD4 T cells, CD8 T cells, B cells, and natural killer cells, often accompanied by exhausted T cells that exhibit diminished proliferative capacity and over-production of pro-inflammatory cytokines [307]. Alongside the use of corticosteroids and other immunomodulatory agents, the defective immunity in COVID-19 patients creates an opportunity for the development of invasive fungal diseases, including PCP [308]. 

## 5. PCP Outbreaks 

Despite the lack of a clearly identified virulence factor, PCP outbreaks occur periodically across the globe. Over the past two decades, there is an increasing trend of PCP outbreaks in immunocompromised patients without HIV/AIDS, particularly in transplant recipients, as recently reviewed by Delliere et al. [309]. The largest number of cases involved in PCP outbreaks was reported in France (150 cases), followed by Australia (97 cases) and the UK (58 cases) (Figure 4, Appendix A [58,77,107,310,311,312,313,314,315,316,317,318,319,320,321,322,323,324,325,326,327,328,329,330,331,332,333,334,335,336,337,338,339,340,341,342,343]).

Types of transplantation associated with PCP outbreaks include both solid organ (kidney, heart, liver, lung, intestine, and pancreas) and hematopoietic stem cell (or bone marrow) transplantation. Outbreaks in solid organ transplant recipients account for approximately 80% of all documented PCP outbreaks [23]. These outbreaks have been documented in over a dozen countries, with the majority of cases occurring in Europe and predominantly affecting renal transplant recipients. 

There has been an intense effort to investigate the epidemiology, contributing factors, and clinical management strategies of these outbreaks [23,109,309]. The transmission mode of PCP outbreaks in transplant patients has been extensively investigated by nearly 30 molecular epidemiological studies, as summarized in Table 3.

In most of these studies, PCP outbreaks are linked to a predominant strain, often even a single strain, of *P. jirovecii* [23,344]. Strikingly, three outbreaks from distant renal transplant centers in Switzerland and Germany were linked to a single *P. jirovecii* strain [345,346]. These observations strongly suggest that the transplant recipients were infected by recent acquisition of *P. jirovecii* through person-to-person transmission. Nevertheless, some studies have demonstrated the presence of coinfections with up to seven *P. jirovecii* strains per patient or multiple distinct *P. jirovecii* strains in the same outbreaks [58,338,340]. 

Despite the relatively high prevalence of these outbreaks, the driving forces behind them remain unknown. The question of whether the outbreaks resulted from the introduction of specific *P. jirovecii* strains with enhanced virulence for transplant patients or from specific conditions that increase patients’ susceptibility to infection, such as certain immunosuppressive or rejection treatment regimens, still needs to be investigated. A better understanding of the infection mechanism and the transmission mode is expected to inform appropriate strategies for clinical management and control of PCP in transplant recipients, as discussed below. 

## 6. Pneumocystis Colonization

*Pneumocystis* colonization is defined as the presence of *Pneumocystis* organisms in the respiratory tract (usually identified by PCR and less frequently by microscopy) without causing symptoms of acute pneumonia [33,347]. There has been increasing effort in addressing the prevalence and significance of *Pneumocystis* colonization, with ~80% (227/282) of PubMed reports related to this subject being published after 2010. 

The reported prevalence of colonization varies greatly among current reports (0–100%), dependent largely on patients’ immune status, underlying health conditions, and ages, as well as the detection methods [33]. In general, immunocompromised patients have a higher colonization prevalence, as high as 68% in HIV-infected patients [348] and 44% in non-HIV patients received corticosteroids, compared to immunocompetent individuals (as high as 24% in adults [349]). In HIV-infected patients, severe immunosuppression (CD4 cell count ≤ 50/μL) and lack of prophylaxis increase the odds of colonization. An exceptionally high prevalence of 100% in immunocompetent individuals was reported from a study of seemingly healthy adults in Chile who died suddenly or violently using nested-PCR and a large weight of autopsied lung tissues [350].

Patients with various chronic lung diseases represent another exception to the immunocompetent population, as studies have reported a high prevalence of *Pneumocystis* colonization in these patients, ranging from 34% to 100%, although the sample size in some of these studies was very small [351,352,353]. 

Furthermore, children, particularly infants, represent a unique exception to the immunocompetent population, as their immune systems are not fully developed, which makes them more susceptible to *Pneumocystis* colonization, with a prevalence ranging from 32% to 100% [164,354,355].

Despite the widespread colonization in both immunocompetent and immunodeficient individuals, many questions remain unanswered regarding its mechanisms and clinical significance. The size of the colonized healthy population is expected to be substantially larger than that of the colonized immunodeficient population; this hypothesis is potentially supported by the immune evasion mechanism associated with the large major surface glycoprotein family, which is thought to be operative in immunocompetent hosts [356,357]. Nonetheless, it remains uncertain if colonization in some cases, especially in the early stage, may represent subclinical or self-limited infection, as studies of *Pneumocystis*-infected healthy animals have demonstrated mild pathological changes in the lungs [358,359]. Distinguishing between subclinical infection and colonization as well as latency discussed below is difficult based on the current data, and it may also depend on how these terms are defined, particularly in terms of the level of damage to the host and the host immune response, as discussed for other pathogens [360]. It also remains undetermined if *Pneumocystis* colonization could play a role in the progression of some chronic pulmonary diseases, particularly COPD, as hypothesized in some studies [33]. It would be necessary to determine whether colonization contributes to these diseases or is a result of certain levels of damage to the lungs due to these diseases.

Another potentially important role of colonization is that colonized patients may serve as reservoirs for transmission, as discussed above (in Section 2). 

## 7. Reactivation of a Latent Infection Versus Acquisition of a New Infection

One of the key questions regarding the pathogenesis of PCP is whether it arises from the reactivation of a latent infection, as initially hypothesized, or from de novo acquisition of a new infection, as postulated more recently. While the reactivation hypothesis is a long-standing one, it is very difficult to prove this hypothesis as it would ideally require determining the genotypes of a strain acquired earlier in life (e.g., during primary infection as an infant) and a subsequent strain causing clinical PCP many years later. In addition, this hypothesis has been challenged by increasing evidence supporting the possibility of de novo infection. First, multiple animal studies have shown that no *Pneumocystis* organism or DNA can be detected in animals after recovery from experimental infection, implying that the host immune response can completely clear *Pneumocystis* without the organism transitioning to a latent state [358,361]. 

Second, several studies of patients with recurrent PCP have found genetically distinct strains between different episodes in the same patients [362,363], supporting the fact that reactivation though other cases in these studies showed identical strains between different episodes, and thereby supporting recent infection for the latter episodes. 

Third, there are ample reports of detection of sulfa resistance-associated gene mutations in the dihydropteroate synthase gene of *P. jirovecii* from patients without prior exposure to sulfa drugs [364,365,366], suggesting de novo infection transmitted from patients with prior sulfa exposure, although the possibility of unidentified sulfa exposure cannot be ruled out. 

Forth, there were over 20 outbreaks of PCP in transplant recipients (including three outbreaks from distant renal transplant centers in Switzerland and Germany) that were linked to a single or a few strains based on molecular typing, as listed in Table 3. This is very strong evidence for de novo infection via person-to-person transmission rather than reactivation of a latent infection. Of note, several studies have identified coinfections with up to seven *P. jirovecii* strains in the same transplant patients [58,338,340]. This may suggest a continuous acquisition of new strains from different infected individuals over time rather than reactivation of multiple latent strains. 

In-depth understanding of the infection mechanisms is expected to inform appropriate strategies for clinical management and prevention of PCP. Theoretically, if reactivation of latent infection is the primary mechanism, it would not be necessary to implement respiratory isolation of PCP patients other than prophylaxis to protect susceptible individuals. If de novo infection is the primary mechanism, it would be important to prevent infected patients from spreading *P. jirovecii* to others, particularly to immunocompromised individuals [23], though all outbreaks would be stopped by widespread prophylaxis. Validation of these postulations is highly challenging due to the difficulty of tracking whether or not patients had a previous infection and, if so, when it occurred. 

## 8. Prevention of PCP 

At present, there is no vaccine for PCP, and respiratory isolation (by avoiding placement in the same room with PCP patients) is only recommended for susceptible patients in healthcare settings per the current USA CDC guidelines [367]. Chemical prophylaxis is highly effective in preventing PCP in high-risk immunocompromised patients without HIV/AIDS, which is similar to HIV/AIDS patients. Guidelines and the consensuses of experts for the prophylaxis of PCP in various immunocompromised conditions in different countries are listed in Table 4. 

The first-line agent for prophylaxis is the traditional trimethoprim-sulfamethoxazole combination (cotrimoxazole). Alternative choices include dapsone, pyrimethamine-leucovorin, atovaquone, and pentamidine. Details on dosages and duration about these drugs are available from the Guidelines listed in Table 4.

## 9. Concluding Remarks

As a life-threatening opportunistic infection, PCP not only incurs significant healthcare costs [7] but also has profound impacts on the health and overall well-being of individuals with weakened immunity worldwide. PCP in non-HIV patients is often more severe and challenging to diagnose than PCP in HIV/AIDS patients. It is important for clinicians and health care workers to be aware of and identify various immunodeficient conditions and risk factors for PCP at an earlier stage in order to initiate timely and efficient prophylaxis or treatment. In addition, given the increasing reports of drug resistance in other fungal and bacterial infections, there is a clear need to increase efforts to investigate the epidemiology and significance of genetic mutations associated with drug resistance in PCP, as advocated recently by the WHO [15].

## Figures and Tables

**Figure 1 jof-09-00812-f001:**
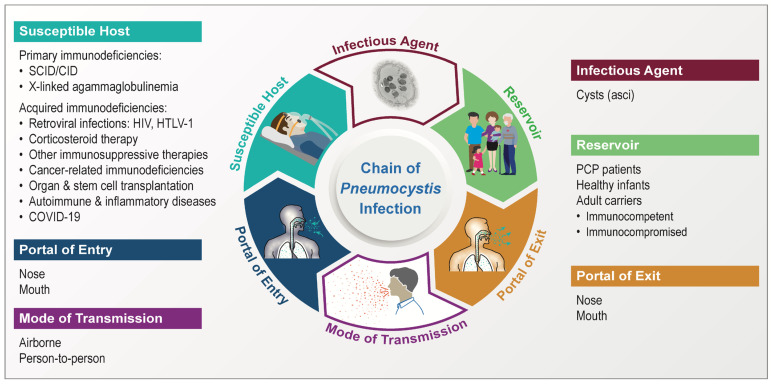
*Pneumocystis* infection chain. The infectious agent is exemplified by the cyst (also known as ascus) of *P. murina* in infected mouse lungs revealed by transmission electron microscopy at a magnification of 5000×. The cyst is characterized by a thick wall with double electron-dense layers enclosing eight intracystic bodies or spores. For the primary immunodeficiencies (under the Susceptible Host), only a limited number of conditions are listed as examples to enhance visual clarity, with more details described in Section 4.5.

**Figure 2 jof-09-00812-f002:**
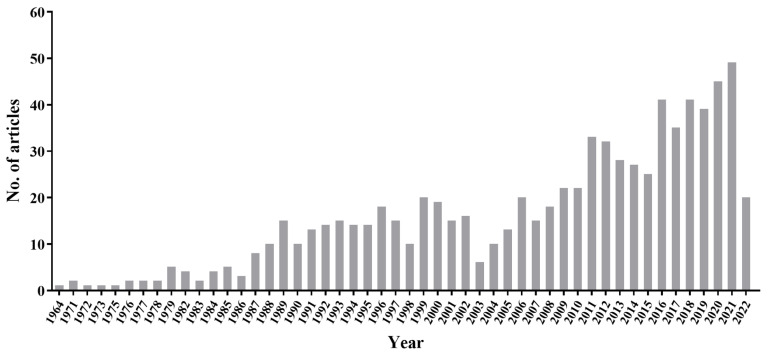
The number of reports on PCP in transplant recipients from 1964 to 2022 based on search of PubMed using keywords “*Pneumocystis* AND Transplant” or “*Pneumocystis* AND Transplantation” in “Title/Abstract”.

**Figure 3 jof-09-00812-f003:**
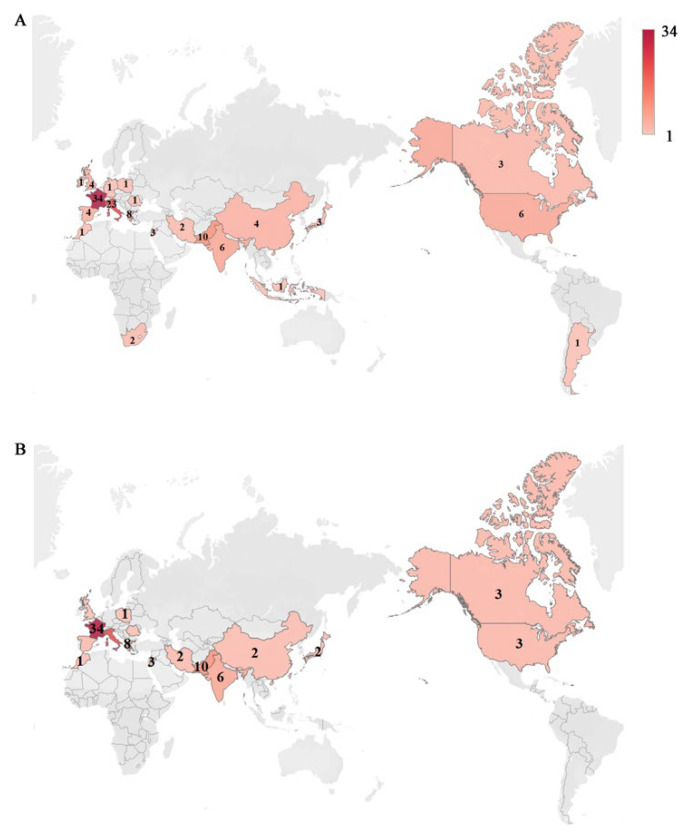
Heatmaps depicting the incidences of co-infection with *P. jirovecii* and SARS-CoV-2. (**A**) The total incidences of co-infection with *P. jirovecii* and SARS-CoV-2 in patients regardless of HIV/AIDS. (**B**) The incidences of co-infection with *P. jirovecii* and SARS-CoV-2 in patients without HIV/AIDS. The number on the map represents the reported number of cases. Details on statistics and related references for all cases are provided in Appendix A.

**Figure 4 jof-09-00812-f004:**
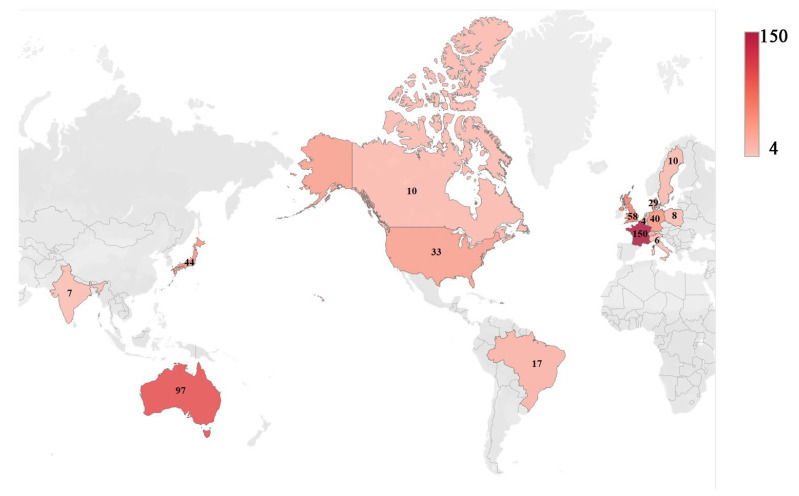
Heatmap depicting the incidences of PCP outbreaks in transplant patients. The number on the map represents the reported number of cases regardless of whether genotyping analysis was conducted. Details on statistics and related references for all cases are provided in Appendix A.

**Table 2 jof-09-00812-t002:** Primary immunodeficiencies associated with PCP.

Immunodeficiency Category (according to the Classification by the Inborn Errors of Immunity Committee 2022 [220])	Genetic Defects
I. Immunodeficiencies affecting cellular and humoral immunity:	
T-B+ SCID	*IL2RG*, *JAK3*, *ADA*
CID generally less profound than SCID	*CARD11*, *CD40*, *CD40LG*, *DOCK8*, *IKBKB*, *IKZF1*, *IL21R*, *MALT1*, *RFXANK*, *ZAP70*
II. CID with associated or syndromic features:	
DNA repair defects other than those in Category I	*DNMT3B*, *ZBTB24*
Hyper IgE syndromes (HIES)	*STAT3*
Defects of vitamin B12 and folate metabolism	*MTHFD1*, *SLC46A1*, *TCN2*
Anhidrotic ectodermodysplasia with immunodeficiency	*IKBKB*, *IKBKG*
Calcium channel defects	*ORAI1*
Other combined immunodeficiencies with syndromic features	*IKZF3*, *KMT2A*, *SKIV2L*, *SP110*
III. Predominantly antibody deficiencies:	
Agammaglobulinemia	*BTK*
Common variable immune deficiency	*NFKB1*
IV. Diseases of immune dysregulation:	
Regulatory T cell defects	*CTLA4*
V. Congenital defects of phagocyte number or function:	
Defects of motility	*CFTR*
Defects of respiratory burst	*CYBB*, *G6PD*
Other non-lymphoid defects	*GATA2*
VI. Defects in intrinsic and innate immunity:	
Predisposition to severe viral infection	*NOS2*
Predisposition to mucocutaneous candidiasis	*IL17RA*, *STAT1*
Other inborn errors of immunity related to leukocytes	*IRF4*
VII. Autoinflammatory disorders	*IFIH1*, *IL36RN*, *TNFRSF1A*
VIII. Complement deficiencies	*C7*
IX. Bone marrow failure	*RTEL1*, *TERC*, *TP53*

Details on genetic defects and references for PCP in patients with the genetic defects shown are provided in Appendix A. This table excludes genetic defects without reports of PCP clinical cases. SCID, severe combined immunodeficiencies; CID, combined Immunodeficiencies; CVID, common variable immune deficiency. T-B+, decreased T cell counts, and normal B cell counts but loss of function.

**Table 3 jof-09-00812-t003:** Published genotyping studies of PCP outbreaks in transplant patients.

First Author [Reference]	Year of Report	Country	No. of Patients	Transplanted Organs	No. of *P. jirovecii* Strains/Clusters
Olsson [312]	2001	Sweden	10	Kidney	3
Hocker [313]	2005	Germany	7	Kidney	5
de Boer [314]	2007	Netherlands	8	Kidney	>5
Schmoldt [315]	2008	Germany	16	Kidney	1
Yazaki [316]	2009	Japan	27	Kidney	1
Arichi [317]	2009	Japan	9	Kidney	5
Gianella [318]	2010	Switzerland	20	Kidney	1
Wynckel [319]	2011	France	17	Kidney	2
Thomas [321]	2011	UK	21 * 11 *	Kidney Kidney	>5 >5
Pliquett [322]	2012	Germany	17	Kidney	3
Brunot [323]	2012	France	7	Kidney	1
Rostvet [326]	2013	Denmark	29	Kidney, liver	3
Debourgogne [327]	2014	France	13	Kidney	2
Gits-Muselli [329]	2015	France	6	Kidney	1
Desoubeaux [330]	2016	France	4	Liver	1
Mulpuru [331]	2016	Canada	10	Kidney	1
Urabe [332]	2016	Japan	8	Kidney	1
Inkster [333]	2017	UK	24	Kidney	2
Vindrios [107]	2017	France	7	Heart	1
Robin [334]	2017	France	12	Stem cell	5
Frealle [335]	2017	France	5	Kidney	2
Alanio [336]	2017	France	13	Kidney	1
		Belgium	5	Kidney	1
		UK	2	Kidney	1
Wintenberger [337] and Charpentier [338]	2017	France	12	Lung, kidney, heart, liver	9
Nevez [339]	2018	France	22	Kidney	1
Ricci [340]	2018	Brazil	17	Kidney	5
Veronese [341]	2018	Italy	6	Heart	2
Szydlowicz [342]	2019	Poland	8	Kidney	>5
Hosseini-Moghaddam [343]	2020	France	10	Heart, kidney, liver	1
Azar [58]	2022	USA	19	Kidney	5

This is a simplified presentation to enhance visualization, with additional details (including duration of outbreaks and genotyping methods) provided in Appendix A. * Involved in two separate outbreaks in different hospitals.

**Table 4 jof-09-00812-t004:** Guidelines and expert consensuses for the prophylaxis of PCP and other opportunistic infections in immunocompromised patients without HIV/AIDS.

Countries	Release Years	Immunocompromised Conditions	References
Asia	2021	Systemic lupus erythematosus	[368]
Australia	2014	Haematological malignancies and stem cell transplantation	[369]
Europe	2016	Haematological malignancies and stem cell transplantation	[52]
Europe	2022	Autoimmune inflammatory rheumatic diseases	[370]
Europe	2018	Relapsed or refractory lymphocytic leukemia patients	[371]
Germany	2021	Haematopoietic stem cell transplantation, solid tumors, and autoimmune disorders	[372]
Japan	2019	Use of methotrexate in rheumatoid arthritis	[373]
Spain	2022	Autoimmune rheumatic diseases	[374]
UK	2017	Solid tumors in children	[375]
UK	2020	Anti-neutrophil cytoplasm antibody-associated vasculitis patients receiving Rituximab for maintenance	[376]
USA	2019	Solid organ transplantation	[111]
USA	2018	Cancers	[377]

## Data Availability

Not applicable.

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
