# Peer review of "Trends in the Epidemiology of Pneumocystis Pneumonia in Immunocompromised Patients without HIV Infection"

_jof, 2023, doi:10.3390/jof9080812_

Round 1

Reviewer 1 Report

This review article was very well written and references thorough and up to date (refs. listed as late as 2022). For the Pneumocystis research community, this manuscript will be a vital paper for referencing past work in the filed and for this outside the field a good synopsis of the Pneumocystis especially in the clinical/hospital setting.  Very well written.

Author Response

Response: We are grateful for the reviewer’s positive comment. His/her kind remarks are truly appreciated.

Reviewer 2 Report

Dear EIC, Prof. Dr. David S. Perlin

Dear AE,

I hope you are doing well.

This is my review result for manuscript #jof-2397561.

This study qualitatively reviewed the risk factors and prevalence of PCP among the non-HIV population. Although there are a lot of papers in this regard, I think this review is a good update to the field. The writing is acceptable, and the manuscript flow seems good. However, I have prepared some minor and major comments to level up the manuscript.

Comments

  • This manuscript lacks some related graphs, especially the geographic prevalence and distribution, which should be depicted by related maps and graphs.
  • Immunological interactions should be depicted by pathway illustration, especially in the COVID-19 section, which has novel content.
  • A meta-analysis study was recently published and investigated the prevalence and risk factors associated with PCP patients among COVID-19 populations. I’m wondering why the authors didn’t use that in their manuscript, especially in the COVID-19 section. So kindly use the results of this study and discuss them in the related section. Below is the link to this paper.

Link:  https://www.apjtm.org/article.asp?issn=1995-7645;year=2022;volume=15;issue=10;spage=431;epage=441;aulast=Khodadadi

Author Response

Comments

  • This manuscript lacks some related graphs, especially the geographic prevalence and distribution, which should be depicted by related maps and graphs.

Response: We greatly appreciate this constructive suggestion and have added two figures depicting the geographic distribution of reported PCP cases in transplant recipients (Fig 4) and COVID-19 patients (Fig 3), which are two main foci of this review. Original data related to these figures are provided in Supplemental Table S3 and Table S4.

  • Immunological interactions should be depicted by pathway illustration, especially in the COVID-19 section, which has novel content.

Response: Since immunological interactions have been well covered in multiple recent papers and do not appear to align closely with the focus of this manuscript, we have chosen to add a section (#4.8) to discuss the effects of various immunodeficiencies on different components of the immune system (pages 18-21, lines 353-417), which we believe is more pertinent to the field of epidemiology. Regarding the immunological interactions, we have provided citations for two recent publications in the revised manuscript (ref #180 and 181).

  • A meta-analysis study was recently published and investigated the prevalence and risk factors associated with PCP patients among COVID-19 populations. I’m wondering why the authors didn’t use that in their manuscript, especially in the COVID-19 section. So kindly use the results of this study and discuss them in the related section. Below is the link to this paper.

Link:  https://www.apjtm.org/article.asp?issn=1995-7645;year=2022;volume=15;issue=10;spage=431;epage=441;aulast=Khodadadi

Response: We sincerely value the reviewer's concern and would like to express our apologies for omitting the citation recommended. While we acknowledge that the reference was included in our earlier draft, we regrettably cannot pinpoint the exact reason for its omission in the final submission. It is possible that we made the decision to exclude the reference in order to reduce the length of the manuscript, especially if it was not indexed in PubMed or if a similar report had already been cited. As requested, we have added the suggested reference along with some comments in the revised version of the manuscript (page 15, line 304 and page 17, lines 329-336).

Reviewer 3 Report

Dear Editor,

The article entitled: "Trends in the Epidemiology of Pneumocystis Pneumonia in Immunocompromised Patients without HIV Infection" by Xue et al is a comprenhensive and very well written review article that focuses on PCP in non-HIV patients. Additionally to this, the review increases awareness and knowledge of diagnosis and treatment of PCP.

To my opinion the manuscript is suitable for publication and there are only some minor comments:

1. Section 2. Authors should specify why infants are more vulnerable to PJ infection in comparison to older children and adults? Is any known reason for this?

2. Section 2. Please specify why cellular immunity is a predisposing factor for PJ infection. I would suggest to add a small paragraph about the implication of immunity in this disease.

3. Table 3 is difficult to read. Please reduced it if possible without loss of significance.

Author Response

  1. Section 2. Authors should specify why infants are more vulnerable to PJ infection in comparison to older children and adults? Is any known reason for this?

Response: We appreciate the reviewer’s questions. To address these questions, we have revised the related paragraph as follows (in page 13, lines 257-262):

“Infants are at a higher risk for infection than healthy older children and adults because their immune systems are not fully developed, with limited or weakened innate and adaptive immune responses to pathogens [140, 141]. Premature birth and malnourishment, along with the use of corticosteroids and other immunosuppressive agents as part of their treatment, can further compromise the immune system and increase their susceptibility to infection, including P. jirovecii infection [139, 142-146].”

In addition, we did more detailed review on primary immunodeficiencies associated with PCP, and provided a comprehensive list of all known primary immunodeficiencies associated with PCP (Table S2) based on the recent classification of primary immunodeficiencies published by the International Union of Immunological Societies Expert Committee (Tangye et al. J Clin Immunol 2022;42:1473).

  1. Section 2. Please specify why cellular immunity is a predisposing factor for PJ infection. I would suggest to add a small paragraph about the implication of immunity in this disease.

Response: We are grateful for the comment and suggestion. Instead of focusing on cellular immunity alone, we have chosen to add a section (#4.8) to discuss the effects of various immunodeficiencies on different components of the immune system (pages 18-21, lines 353-417), which we believe is more pertinent to the field of epidemiology.

  1. Table 3 is difficult to read. Please reduced it if possible without loss of significance.

Response: As requested, we have reformatted the table by removing two columns, with the hope of improving visualization. All data in the original table are included in Supplemental Table S3, which includes additional reports of PCP outbreaks in transplant recipients without genotyping analysis and sporadic PCP cases in transplant recipients used in the heatmaps to the geographic distributions of PCP cases as requested by Reviewer 1.

Round 2

Reviewer 2 Report

I convinced by response to my comments.